# Drug Distribution and Penetration of Foam-Based Intraperitoneal Chemotherapy (FBIC)

**DOI:** 10.3390/ph16101393

**Published:** 2023-10-02

**Authors:** Carolina Khosrawipour, Jakub Nicpoń, Zdzisław Kiełbowicz, Przemysław Prządka, Bartłomiej Liszka, Veria Khosrawipour, Said Al-Jundi, Shiri Li, Hien Lau, Joanna Kulas, Piotr Kuropka, Agata Diakun, Wojciech Kielan, Mariusz Chabowski, Agata Mikolajczyk-Martinez

**Affiliations:** 1Faculty of Medicine, Wroclaw Medical University, 50-367 Wroclaw, Poland; 2Department and Clinic of Surgery, Faculty of Veterinary Medicine, Wroclaw University of Environmental and Life Sciences, 50-366 Wroclaw, Poland; 3Department of Surgery, Petrus-Hospital Wuppertal, Teaching—Hospital of the University of Medicine Dusseldorf, 42283 Wuppertal, Germany; 4Division of Colon and Rectal Surgery, Department of Surgery, New York Presbyterian Hospital-Weill Cornell College of Medicine, New York, NY 10065, USA; 5Department of Surgery, University of California Irvine (UCI)—Medical Center, Irvine, CA 92868, USA; 6Faculty of Veterinary Medicine, Wroclaw University of Environmental and Life Sciences, 50-375 Wroclaw, Poland; 107089@student.upwr.edu.pl; 7Department of Biostructure and Animal Physiology, Wroclaw University of Environmental and Life Sciences, 51-631 Wroclaw, Poland; 82nd Department of General Surgery and Surgical Oncology, Wroclaw Medical University, 50-556 Wroclaw, Poland; agata.diakun@umw.edu.pl (A.D.);; 9Faculty of Medicine, University of Science and Technology Wroclaw, 58-376 Wroclaw, Poland; 10Department of Surgery, 4th Military Hospital, 50-981 Wroclaw, Poland; 11Department of Biochemistry and Molecular Biology, Faculty of Veterinary Medicine, Wroclaw University of Environmental and Life Sciences, 50-375 Wroclaw, Poland

**Keywords:** foam-based intraperitoneal chemotherapy, intraperitoneal chemotherapy, oxaliplatin, doxorubicin, peritoneal metastasis

## Abstract

For decades, intraperitoneal chemotherapy (IPC) was used as a liquid solution for the treatment of peritoneal metastasis. Due to its advantageous physical properties, foam-based intraperitoneal chemotherapy (FBIC) was recently proposed as a treatment for peritoneal metastasis. For the first time, this study intends to examine the feasibility, expansion, drug distribution, and penetration of FBIC in vivo. Three swine received contrast-enhanced FBIC doxorubicin delivered using a bicarbonate carrier system. During the procedure, intraoperative blood analyses and periumbilical diameter, as well as foam distribution, penetration, and expansion of the FBIC were analyzed. The swine received an abdominal CT scan to evaluate the contrast distribution. Furthermore, a hematoxylin-eosin (HE) staining of peritoneal samples was performed, and fluorescence microscopy was conducted. FBIC was performed without complications. The periumbilical diameter peaked after 5 min and then decreased. Blood analyses showed changes in blood parameters, with a reduction in the pH levels of serum calcium and potassium. CT scan detected contrast-enhanced FBIC throughout the abdominal cavity. Fluorescence microscopy confirmed that all areas were exposed to doxorubicin and no pathologies were detected in the HE histology. Our preliminary results are quite encouraging and indicate that FBIC is a feasible approach. However, in order to discuss possible clinical applications, further studies are required to investigate the pharmacologic, pharmacodynamic, and physical properties of FBIC.

## 1. Introduction

Clinical therapy for peritoneal surface malignancies is associated with many challenges and complications. As of now, it remains difficult to slow down the continuous metastatic spread of primary and secondary malignancies within the peritoneal cavity. This, along with complications arising from the metastatic spread, causes an overall bad prognosis for peritoneal metastasis (PM) [1,2,3]. There has been an attempt to manage PM via locoregional anticancer drug installation in multiple forms and combinations, with and without additional surgical procedures. Many scientists focus on improving the outcome of treatment of peritoneal surface malignancies via molecular biological research [4,5], pharmacological enhancements [6,7], or improving and evaluating more clinically established concepts like HIPEC and cytoreductive surgery [8,9]. For decades, intraperitoneal chemotherapy (IPC) was administered into the abdominal cavity as a liquid solution. Recently, pressurized intraperitoneal aerosol chemotherapy (PIPAC) has become an alternative method to treat PM due to the enhanced physical properties [10,11,12] of aerosolized fluid chemo solution. For the first time, we aim to evaluate foam-based intraperitoneal chemotherapy (FBIC) as a potential new drug carrier for in vivo IPC delivery. The concept of foam as a carrier substance was proposed in 2020 and evaluated by Schubert et al. [13]. However, no in vivo study on intraperitoneal foam delivery has yet been conducted. The studied concept of FBIC points out a variety of potential benefits in the delivery of chemotherapy dissolved in a foam medium. Foam exhibits some physical characteristics that could be advantageous, which would potentially surpass the limitations set by both aerosol and liquid applications [13]. For example, the surface contact of a concentrated drug can be expanded using foam without diluting the drug concentrations [13]. Additionally, the gradual degradation of the foam enables extended drug contact in the peritoneum. Another aspect is that foam is less affected by gravitational effects than a dense liquid medium. This indicates that a more multidirectional expansion of the foam is possible. These improvements and advantages are significant. However, foam is far more complex than a liquid medium and needs to be studied intensely before a clinical application is feasible. In order to evaluate whether FBIC is a possible approach for PM treatment, we investigated its applicational feasibility, foam expansion, and drug distribution in an experimental in vivo setting. During the intervention, we focused on changes affecting the pH, pCO_2_, pO_2,_ and electrolyte levels. We aimed to evaluate abdominal expansion via the periumbilical diameter and the intra-abdominal distribution of foam. Furthermore, we evaluated whether a CT scan can visualize the foam and its distribution. Finally, we performed a histopathological evaluation of potential changes in the peritoneum and measured the drug penetration rates into the tissue as in similar prior studies [14,15,16]. This study was part of a multistage study in an in vivo swine model. Alongside the aims of the study, we first gathered in vivo data on a variety of important aspects of this potential new approach, including surgical parameters, gasometrical data, foam stability, and the first postoperative histological evaluation.

## 2. Results

### 2.1. Feasibility and General Safety Concerns Regarding the Intraperitoneal Foam Application

The in vivo experiments on the three swine were successfully conducted (Figure 1). No intraoperative or postoperative complications were detected. All animals survived the surgery and the following postsurgical recovery. No major complications were noted. No postoperative macroscopical changes were observed on day seven after the final cadaver autopsy. No major applicational problems were observed during the operation. A wide range of data were collected, and no major anesthesiology problems or issues during recovery were detected. All the swine drank and ate adequately after the procedure. No pain or behavioral changes were seen during recovery. 

### 2.2. Abdominal Extension and Intraoperative Blood Gas Analyses

Diameters were measured at each stage of the procedure for all three swine. The initial periumbilical diameters varied from around 68 cm to 74 cm (Figure 2). During the insufflation of CO_2_, the periumbilical diameter increased. The applied insufflation pressure was between 12 and 14 mmgH. After foam insufflation, the periumbilical diameter increased and surpassed the values noted prior to CO_2_ insufflation. The foam insufflation was discontinued when the diameter reached a maximum of 84 cm. Five minutes after the foam insufflation, the mean periumbilical diameter continuously decreased. At the end of the procedure, the periumbilical diameters were 77 cm, 78 cm, and 79 cm, which were above the initial diameters. Fifteen min into the procedure, a laparoscopic camera was introduced into the abdomen. Laparoscopically, a partial collapse of the foam was observed. Nevertheless, the foam still appeared to cover the peritoneal surface.

### 2.3. Development of Intraoperative Gasometric Parameters and Electrolytes

Arterial blood gas analyses were conducted for all swine before starting the treatment as well as 30 min into the treatment. The PH decreased 30 min into the procedure. The drop ranged from a pH of 7.23 ± 0.027 to 7.1 ± 0.03 (Figure 3). At the same time, an increase in the pCO_2_ (*p* > 0.05) and a decrease in the pO_2_ (*p* < 0.05) were noted. Additionally, a variety of changes in the electrolytes were registered.

In the beginning, serum sodium was 145.3 mmol/L ± 2.3 (0 min) and it increased to 146.3 mmol/L ± 2.08 (30 min). However, serum potassium decreased significantly after 30 min with 3.4 mmol/L ± 0.1 versus 3.87 mmol/L ± 0.058 (*p* < 0.05) at 0 min. Also, serum calcium was reduced significantly after the initial levels of 1.42 mmol/L ± 0.117 versus 0.93 mmol/L ± 0.168 (*p* < 0.05) after 30 min.

### 2.4. Evaluation of Computer Tomography

Intraoperative foam delivery was visualized successfully via CT (Figure 4). Horizontal, vertical, and transversal foam expansions were detected. Trapped residual air was seen around the entrance of the trocar. Small amounts of contrast-enhanced foam fluid were detected in multiple areas within the abdominal cavity. Some accumulation of contrast media was seen within the deeper spots of the abdominal cavity.

### 2.5. Evaluation of Histology

After euthanization and autopsy of swine according to established protocols [17,18], tissue samples were removed from multiple places in the abdominal cavity. These tissue samples were retrieved from different sites in the parietal and visceral peritoneum. HE histology did not reveal any pathologies (Figure 5). No indication of white blood cell infiltration was detected. No signs of tissue degradation or necrosis were observed. Following fluorescent microscopy, tissue probes revealed that all peritoneal locations were in contact with doxorubicin. The mean depth of doxorubicin penetration at the parietal peritoneum (upper abdomen) was 240 ± 49 µm; at the parietal peritoneum (lower abdomen), it was 273 ± 57 µm; and at the small intestine, it was 330 ± 85 µm, D: stomach 188 ± 62 µm.

## 3. Discussion

For the first time, an in vivo foam application into the peritoneal cavity was successfully carried out. Surgical intervention and foam delivery were performed without any surgical or technical problems. We witnessed none of the technical obstacles that were previously considered to be a major problem. However, some technical challenges still need to be addressed. One such major aspect was the quantification of the applied total foam volume. With an external foam delivery device, the total volume of the applied foam could not be quantified or estimated. Thus, the periumbilical diameter had to be used in an indirect way to evaluate abdominal expansion and limiting foam insufflation. Also, vital parameters and inspiration pressure from the mechanical ventilation system were analyzed. Measuring the periumbilical diameter has previously been used in different studies that are focused on intraabdominal pressure [19,20,21]. The provided data indicate that the diameter at the peak of foam delivery surpassed the diameter during diagnostic laparoscopy. However, no data were available on the actual intra-abdominal pressure at the peak of foam insufflation. Only indirect signs of clinically relevant intra-abdominal pressure were detectable, such as an increase in blood pressure and a simultaneous drop in heart rate. It is disputable whether the buildup of pressure is only due to foam and is not a result of serum electrolyte imbalances. During laparoscopic gas insufflation, similar observations on vital parameters were observed [22,23,24,25]. Therefore, this aspect is not completely new. However, it is noteworthy that there was a continuous and steady decrease in the periumbilical diameter. This could be an indication of rapid CO_2_ absorption from the abdominal cavity. It is important to note that the decrease in the periumbilical diameter allows the swine to be extubated after the procedure. This indicates that the mechanical respiratory functions are not limited by the levels of intraperitoneal foam. This discovery is significant because prior to our experiments, a major concern was that the intra-abdominal expansion might interfere with the movement of the diaphragm [26,27]. Another interesting observation was that the rate of foam disintegration did not seem to correspond with the decrease in the abdominal cavity size. At around 15–20 min into the procedure, the abdominal cavity could be visualized laparoscopically via the optical systems. The disintegration of the foam was faster than the CO_2_ absorption. Therefore, large pockets of air make visual imaging possible. Visual imaging detects foam at various locations in the abdominal cavity, especially along the abdominal wall. This is of importance because the abdominal wall acts as a so-called “dome”. This “dome” is the uppermost part of the peritoneum and is barely accessible for most medications. The continuous reabsorption of CO_2_ is why the periumbilical diameter decreases. This is the most likely reason for the absence of an accidental leak in the trocar within the abdominal cavity. In our in vivo experiments, no leaks were observed. Therefore, the reabsorption of CO_2_ is the most probable explanation. Even if the foam were to collapse on its own, CO_2_ would still be trapped in the abdominal cavity. The drop in serum pH could be explained by an increase in serum CO_2_. This increase in CO_2_ may also influence serum potassium levels, which later decrease. This is not surprising as lower levels of potassium are regularly observed with decreased pH levels [28,29,30]. Concerning other electrolytes, the serum sodium level seemed to be unaffected while the serum calcium level was significantly reduced. This is explained by the citrate component of the intraperitoneal foam, which is known to react with the calcium in the blood [31,32]. This phenomenon could be a potentially independent observation and might not be related to the secondary effects of the serum pH. Therefore, the role of citric acid and bicarbonate in this carrier system needs to be further explored. Other potential side effects of bicarbonate administration and exposure are well known. The adverse effects [33,34] following the administration of citric acid and its sodium salts in “toxic” or higher quantities are related to a typical serum calcium deficiency. Furthermore, no postoperative wound complications were seen, which may have been associated with an increased intra-abdominal pressure of the applied substance. This is important as a variety of factors may be associated with periumbilical wound infections and fistula formation [35].

Further studies need to be conducted to provide sufficient evidence regarding safety, technical application, and biocompatibility.

## 4. Materials and Methods

### 4.1. Sequence of Procedures 

A laparoscopy was performed on each swine (Polish white landrace breed). The swine were prepared accordingly for total anesthesia and intubation. FBIC was introduced into the abdominal cavity. The last swine in the queue received an additional dosage of foam-based intrathoracic chemotherapy in the right hemithorax. After the procedure, the swine were observed for 7 postoperative days. Euthanization was performed on day 7. Following the euthanization, the FBIC was reintroduced into the abdominal cavity. An abdominal CT scan was performed. Afterwards, a median laparotomy was performed, and the cavity was inspected. Tissue samples were removed from multiple intraabdominal locations and the tissue was further analyzed according to established protocols [14,15]

### 4.2. The Laparoscopic In Vivo Swine Model

This study included three 65-day-old swine. A standard laparoscopic approach was used for the procedure, and the swine were prepared for a laparoscopic setting [16,17]. Swine were premedicated with an intramuscular injection of midazolam (0.3 mg/kg, WZF Polfa S.A., Warsaw, Poland), medetomidine (0.02 mg/kg, Cepetor 1 mg/mL, CP-Pharma Handelsgesellschaft, Burgdorf, Germany), and ketamine (9 mg/kg, Ketamine 100 mg/mL, Biowet Puławy sp. z o.o., Pulawy, Poland) mixture. Analgesia was performed with propofol at 1 mg/kg. The swine were intubated, and further anesthesia was continued with isoflurane 1%. Additional analgesia was provided with fentanyl 2 µg/kg and crystalloid fluid at 0.2–0.3 µg/kg/min. The swine were placed in a supine position. An infra-umbilical mini laparotomy was performed, and another was performed about 8 cm away from the first one. A 10 mm trocar (Kii^®^Balloon Blunt Tip System, Applied Medical, Rancho Santa Margarita, CA, USA) was inserted through the infra-umbilical trocar while a 5 mm trocar was inserted at the other site (Figure 1A). The abdominal cavity was insufflated with CO_2_ to maintain a capnoperitoneum (Olympus UHI-3 insufflator, Olympus medical life science and industrial divisions, Olympus, Shinjuku, Tokyo Japan). An initial diagnostic check-up was made via laparoscopic imaging via a 5 mm camera system (Karl Storz 5 mm/30° Laparoscope/Tuttlingen, Germany). After visual confirmation of no anomalies, the “foam-insufflation” tube of the foam generating system was introduced into the 10 mm trocar. The correct placement was confirmed via laparoscopic visual imaging. Then, the laparoscope was removed, a temperature probe was inserted through the trocar, and the CO_2_ from the capnoperitoneum was evacuated. Another temperature probe was placed on the abdomen from the outside and fixed there using adhesive tape.

### 4.3. Euthanization

The swine were premedicated with an intramuscular injection of midazolam (0.1 mg/kg, Midanium 5 mg/mL, WZF Polfa S.A., Poland), medetomidine (0.02 mg/kg, Cepetor 1 mg/mL, CP-Pharma Handelsgesellschaft, Germany), and ketamine (8 mg/kg, Ketamina 100 mg/mL, Biowet Puławy sp. z o.o., Poland) mixture. Then, they were euthanized according to recommendations [18] with an intravenous injection of Sodium Pentobarbital with Pentobarbital (50 mg/kg with 12 mg/kg, Morbital 133.3 mg/mL + 26.7 mg/mL, Biowet Pulawy Sp. z o.o., Poland). The postmortem swine cadavers were placed in a supine position. FBIC was applied to the abdominal cavity. This was performed according to the in vivo protocol using doxorubicin and contrast media. After delivery of FBIC, the swine were placed into the CT for an abdominal CT scan. Following the scan, a median laparotomy was performed, and tissue samples were removed at several locations for further histological analysis via light and fluorescence microscopy. A thorough exploration of the abdominal cavity was conducted.

### 4.4. Histopathological Examination via Light Microscopy

Samples for light microscopy were fixed for 48 h in 10% neutral buffered formalin (Fisher Scientific, Hampton, VT, USA). Formalin-fixed samples were prepared for paraffin processing via serial dehydration in increasing concentrations of ethanol solutions using a tissue processor (Leica TP1020, Leica Microsystems, Wetzlar, Germany). After preparation, tissues were embedded in paraffin wax using a tissue embedder (Leica EG 1150C, Leica Microsystems, Wetzlar, Germany). Paraffin-embedded tissue blocks were sectioned into 5 µm sections using a microtome (Leica RM 2255, Leica Microsystems, Wetzlar, Germany). Five µm sections were stained with HE. All slides were imaged using an inverted microscope (Nikon Ti-E Widefield microscope, Nikon Instruments Inc., Tokyo, Japan).

### 4.5. Histological Examination via Fluorescence Microscopy

After treatment, all intraperitoneal tissue samples were rinsed with sterile NaCl 0.9% solution to eliminate superficial cytostatic agents and immediately frozen in liquid nitrogen for fluorescence microscopy. Cryosections (7 µm) were prepared from different areas of the specimen. Sections were mounted with VectaShield containing 1.5 µg/mL 4′,6-diamidino-2-phenylindole (ProLong^®^ Gold Antifade Reagent with DAPI, Thermo Fisher Scientific, Hampton, VT, USA) to stain nuclei. The penetration depth of doxorubicin was measured using a Nikon Eclipse 80i fluorescence microscope (Nikon Instruments Europe B.V. Amsterdam, The Netherlands). The distance between the peritoneal surface and the innermost positive staining for doxorubicin accumulation was reported in micrometers.

### 4.6. The Bicarbonate-Based Foam Carrier

The major components of the foam were citric acid (Sigma-Aldrich, St. Louis, MO, USA) and sodium bicarbonate (Sigma-Aldrich, St. Louis, MO, USA). As a chemotherapeutic component, doxorubicin hydrochloride (1.5 mg/m^2^ body surface, PFS^®^, 2 mg/mL, Pfizer, Sandwich, UK) was added. Furthermore, iodide-based contrast media was introduced for the CT (AccupaqueTM 350 mg J/mL, GE Healthcare, Chicago, USA).

### 4.7. Statistical Analyses of Data

The statistical analyses were performed using GraphPad Prism (Version 8.0.2 (263), Insight Partners, New York, NY, USA). Student *t*-test was used to compare independent groups. The Student *t*-test (parametric test) was used based on the assumption of normal distribution (ND) of the gasometric data. The distribution was evaluated using ND-Histogramm. A quantile–quantile plot was not applied. ND for serum blood parameters has been used commonly and has been scientifically established [36]. Descriptive statistics included the mean, median, and percentiles. Probability (*p*) values were considered as follows: * *p* < 0.05 ** *p* < 0.005, and # *p* > 0.05, with *p*-value < 0.05 considered to be statistically significant.

### 4.8. Ethical Approval and Regulations

Approval of the Local Board on Animal Care according to Polish regulations and European Union law was obtained for the experiments (UCHWALA NR 029/2021/P1).

### 4.9. Graphic Design

For the graphics provided, multiple graphic programs were used. Among these programs were Inkscape 1.0.1,2020, GNU, USA, and programs provided by Windows Office 2019, Microsoft.

## 5. Conclusions

Even though a limited number of in vivo experiments were conducted, we observed that the technical and applicational concepts of FBIC are possible. The limitations that exist are more likely due to the carrier system than the actual delivery of the intraperitoneal foam. These concerns include electrolyte imbalances, which need to be closely monitored and studied in the future. Concerning the local effects of FBIC, we noticed that no adhesions, intraperitoneal complications, or microscopical tissue reactions were detected. In fact, shortly after the FBIC application, we performed a control laparoscopy. However, we faced challenges in technical foam delivery as well as the evaluation and measurement of the total inflow volume. The results of this study are promising, but further research on FBIC is needed to provide more data on this novel treatment concept for PM.

## Figures and Tables

**Figure 1 pharmaceuticals-16-01393-f001:**
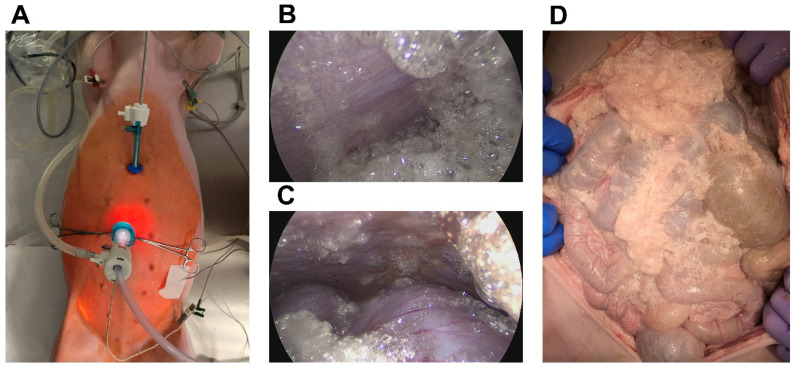
(**A**) Experimental view of in vivo swine model receiving FBIC. Disinfected operative field is colored red (iodine). Surgical entrance sites are visible: a large 10 mm trocar is placed periumbilically and a small 5 mm trocar is placed epigastrally. (**B**,**C**) In vivo laparoscopic view at around 20 min into the surgical procedure. (**D**) Postmortem laparotomic view after FBIC.

**Figure 2 pharmaceuticals-16-01393-f002:**
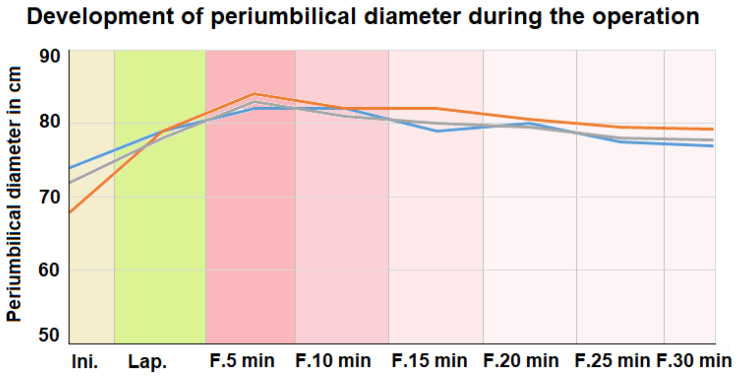
Abdominal expansion of swine abdomen at different stages. (Ini.) Initial diameter before the procedure. (Lap.) Diameter during laparoscopy for diagnostic purposes under 12–14 mmHg. (F.) Diameter after foam insufflation 5 to 30 min into the procedure.

**Figure 3 pharmaceuticals-16-01393-f003:**
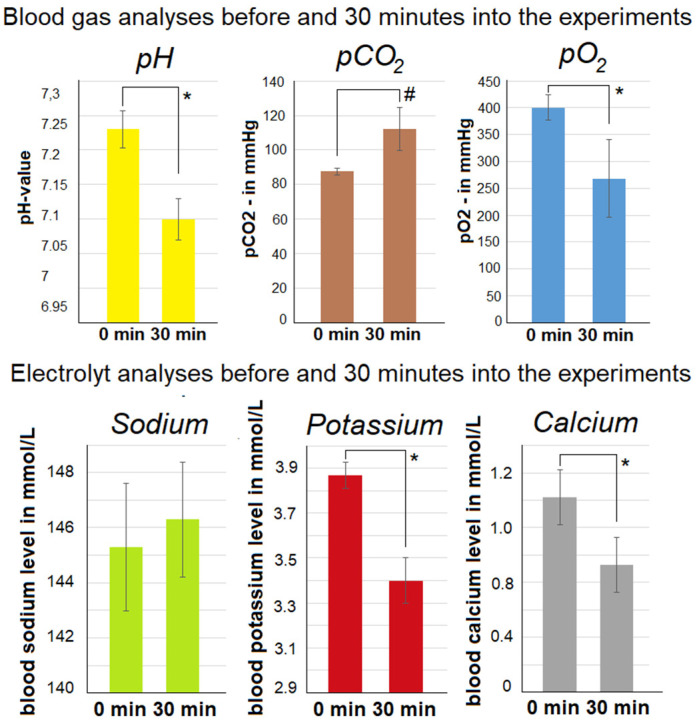
Blood gas analyses during surgery. Blood samples were taken just before foam application and 30 min into foam application. Mean and standard deviation of the measured pH level, pCO_2_ level, and pO_2_ levels, as well as major electrolytes (Na^+^), potassium (K^+^), and calcium (Ca^2+^) levels, were indicated. Significant levels were defined as * = *p* < 0.05 and # = *p* > 0.05.

**Figure 4 pharmaceuticals-16-01393-f004:**
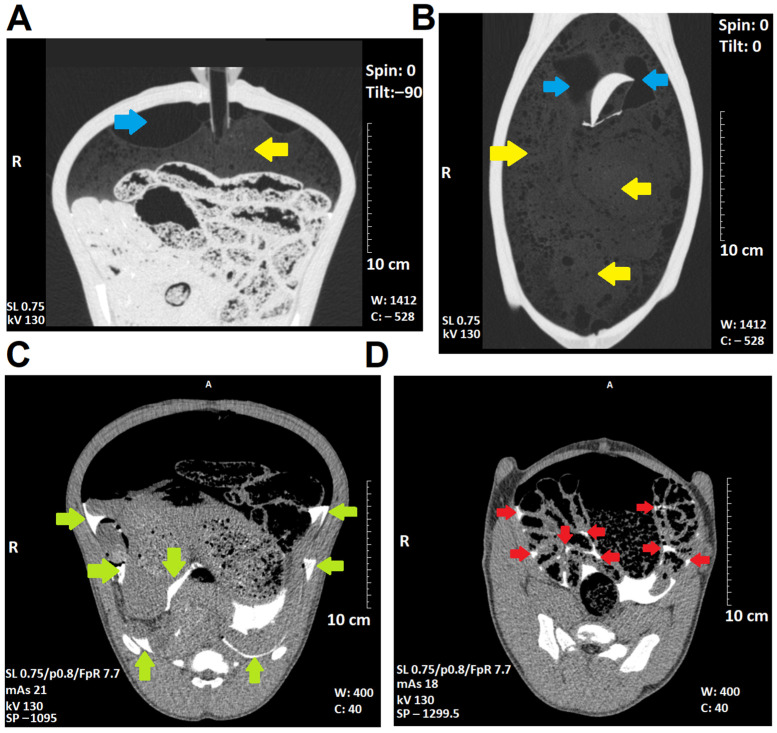
Postmortem foam distribution of FBIC using CT-scan. (**A**) Transversal scan and (**B**) horizontal scan with blue arrows marking trapped residual air around the entrance of the trocar and yellow arrows marking foam (enhanced via contrast media). (**C**,**D**) Transversal scan of the upper and lower abdominal cavity. Accumulation of degraded foam fluid enhanced with contrast media. Green and red arrows indicate the extent of the overall distribution of foam.

**Figure 5 pharmaceuticals-16-01393-f005:**
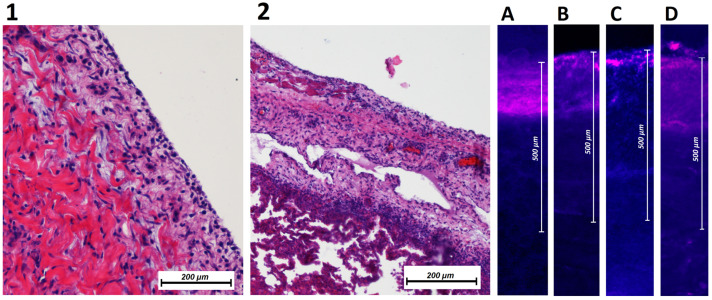
Left side: Histopathology of peritoneal tissue from the (**A**) parietal peritoneum/abdominal wall and the (**B**) visceral peritoneum/small intestine. No specific changes are detectable. No disruption of the peritoneal surface is observed. Right side: Fluorescence histology of multiple abdominal locations. Visualization of doxorubicin (red/purple areas) in the peritoneal tissue. Locations: (**A**): parietal peritoneum (upper abdomen); (**B**): parietal peritoneum (lower abdomen); (**C**): small intestine (**D**): stomach.

## Data Availability

Data is contained within the article.

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
