# Peer review of "Drug Distribution and Penetration of Foam-Based Intraperitoneal Chemotherapy (FBIC)"

_pharmaceuticals, 2023, doi:10.3390/ph16101393_

Round 1

Reviewer 1 Report

well written

please define aim

please define statisticla method more.

( "Factors associated with the occurrence and healing of umbilical pilonidal sinus: a rare clinical entity." Advances in Skin & Wound Care 35.8 (2022): 1-4.) suggested study for the references

Author Response

  1. please define aim

Thank you for this critique. We will further emphasize the aim of the study in the Introduction section.

“During the intervention we will focus on changes affecting the pH, pCO2, pO2 and electrolyte levels. We aim to evaluate the abdominal expansion via periumbilical diameter and the intraabdominal distribution of foam. Furthermore we will evaluate whether a CT-scan can visualize the foam and its distribution. Finally, we will do a histopathological evaluation of potential changes on the peritoneum and a measurement of the drug penetration rates into the tissue.”

  1. please define statisticla method more.

Thanks for this comment we will further define the statical methods. We will add this in the statistic section.

“A student t-test (parametric test) was used based on the assumption of normal distribution (ND) of gasometric data. Distribution was prior evaluated via ND-Histogramm. A Quantile-Quantile plot was not applied. ND for serum blood parameters has been used commonly and have been scientifically established [36].”

Arzideh F, Brandhorst G, Gurr E, Hinsch W, Hoff T, Roggenbuck L, Rothe G, Schumann G, Wolters B, Wosniok W, Haeckel R. An improved indirect approach for determining reference limits from intra-laboratory data bases exemplified by concentrations of electrolytes / Ein verbesserter indirekter Ansatz zur Bestimmung von Referenzgrenzen mittels intra-laboratorieller Datensätze am Beispiel von Elektrolyt-Konzentrationen. Journal of Laboratory Medicine. 2009;33(2): 52-66. https://doi.org/10.1515/JLM.2009.015

  1. ( "Factors associated with the occurrence and healing of umbilical pilonidal sinus: a rare clinical entity." Advances in Skin & Wound Care8 (2022): 1-4.) suggested study for the references

Thank you we will add this to the references as No [35]]

Reviewer 2 Report

This is a novel approach, I am a bit disappointed about the depth of Dox penetration but it is still interesting.

This is novel and interesting

Author Response

Reviewer_2

This is a novel approach, I am a bit disappointed about the depth of Dox penetration but it is still interesting.

Thank you for your comment. The depth is currently limited, however with time we hope to improve the overall foam performance. We believe that the limited penetration is indirectly due to the application device and not because of the methodology. This is currently under evaluation.

Reviewer 3 Report

First of all, I would like to thank MDPI for the possibility of being able to review this work.

This study investigates foam-based intraperitoneal chemotherapy (FBIC) as an alternative to liquid intraperitoneal chemotherapy (IPC) for treating peritoneal metastasis. The research examines FBIC's in-vivo feasibility, expansion, drug distribution, and penetration. Using a bicarbonate carrier-system, doxorubicin-infused FBIC was administered to swine, analyzing foam distribution, penetration, and expansion. Results showed successful FBIC implementation, fluctuation in blood parameters, contrast distribution through CT scans, and uniform doxorubicin exposure in microscopy. Encouraging initial findings highlight FBIC's potential, requiring further research to explore pharmacological and physical aspects before clinical consideration.

This is an interesting and current topic as various alternatives to HIPEC have been proposed.

The title is clear and concise, effectively reflecting the content of the study.

The abstract effectively summarizes the premise; however, it lacks information on specific data and outcomes. Improvement is needed.

The paper's organization is chaotic. It transitions directly from introduction to results without explaining the materials and methods. While materials and methods are presented later, they omit a portion of the methodology. It is crucial that these aspects be revised.

"In the results section, as there is no methodology, there is no specification regarding the surgical procedure, the duration of animal housing, the timing of euthanasia, the animal species involved, etc."

To be candid, this work should be reviewed by its authors before proceeding with peer review. There are numerous aspects that require improvement. I suggest it be returned to them for revision and reworking.

While the study is intriguing, its current structure renders it unpublishable and difficult to comprehend. I recommend a thorough English revision and a complete reorganization of the paper.

The English in the entire paper needs correction. For example, 'instillation' should be used instead of 'installation' in line 60." Where instead of were, line 90. Etc.

Author Response

  1. The abstract effectively summarizes the premise; however, it lacks information on specific data and outcomes. Improvement is needed.

Thank you. In this format the word-count of the abstract is limited and could therefore not include more information without canceling other aspects. This was a compromise between specificity and completeness.

  1. The paper's organization is chaotic. It transitions directly from introduction to results without explaining the materials and methods. While materials and methods are presented later, they omit a portion of the methodology. It is crucial that these aspects be revised.

Thank you for your comment. Indeed it is confusing. We agree. However this is also a pregiven by the journal. The sections are organized by the journal like this. To address this we have already placed a short sequence of procedures in 4.1 so it is easier to get a fast overview.

  1. "In the results section, as there is no methodology, there is no specification regarding the surgical procedure, the duration of animal housing, the timing of euthanasia, the animal species involved, etc."

Thank you for this comment. We can fully understand your frustration with the sequence of Parts: Introduction, Results, and Methods at the later point. However, this in not changeable as this is the journals official sequence. We will request the journal to consider changing this as it is indeed not suitable in our opinion. We have further given more precise details on animal species, timing of euthanasia. The surgical procedure has been described in the method section in a separate chapter: 4.2 The laparoscopic in-vivo swine model.

4. To be candid, this work should be reviewed by its authors before proceeding with peer review. There are numerous aspects that require improvement. I suggest it be returned to them for revision and reworking.

While the study is intriguing, its current structure renders it unpublishable and difficult to comprehend. I recommend a thorough English revision and a complete reorganization of the paper.

Thank you. The structure was not chosen by authors it was determined by the journal according to the journal policies to first present results and methodology at the end. We have added an English- Revision of the manuscript.

Reviewer 4 Report

The article submitted for review studied the feasibility, expansion and penetration of foam based intraperitoneal chemotherapy in a swine model. Though the foam-based method described by the authors has advantages over the current available methods, it is still in its infancy and requires rigorous investigation before its usage in the clinics. The authors reported a variation in the blood parameters and a tissue penetration of ~250 mm following foam injection. Following are the comments concerning this article.

1.      The method of foam generation described is not clear. An illustration showing the setup and foam generation including probes and biopsy sampling areas would be beneficial for the readers.

2.      The authors mentioned that FBIC was reintroduced in the abdominal cavity following euthanization. Could the authors explain the reason for this reintroduction after euthanization?

3.      The dose of DOX used in the study is 1.5mg/m2 bs. How was the dose chosen in this study? What was the dose or the amount of DOX used to create foam after its delivery.

4.      Of other foam carriers what was the rationale to choose bicarbonate-based foam carrier in this study?

5.      Blood parameters were measured after 30mins. Did authors measure blood parameters at any other time point like study midpoint?  

6.      The abdominal expansion was measured in this article. Did the authors measure the kinetics of foam disintegration and time it took for complete foam disintegration?

7.      Throughout the article the decimals are commas instead of dots.

8.      The authors did a good work in discussing their data well and highlighting the challenges of the study.

The quality of English language is sufficient.

Author Response

  1. The method of foam generation described is not clear. An illustration showing the setup and foam generation including probes and biopsy sampling areas would be beneficial for the readers.

Good point. Indeed, the method of foam generation is not meant to be part of this article. The foam creation is a separate issue and does not concern the application because it is already created ex-vivo and just introduced into the abdomen in the final step. The creation of the foam is extensive and will be separately presented in a manuscript. Currently, the major question is whether it is biologically compatible.

  1. The authors mentioned that FBIC was reintroduced in the abdominal cavity following euthanization. Could the authors explain the reason for this reintroduction after euthanization?

Thank you for this valuable and important question. The reintroduction at the time of Euthanization was needed to be able to perform the tissue penetration measurements. The in-vivo measurements of tissue depth cannot be performed because of multiple reasons. Just to mention 2 major reasons: no tissue could be removed from small intestine (Perforation) in an in-vivo model and visibility and intervention is restricted for safe removal of tissue.

  1. The dose of DOX used in the study is 1.5mg/m2 bs. How was the dose chosen in this study? What was the dose or the amount of DOX used to create foam after its delivery.

The dose was 3 mg (1,5mg/m2) which is the corresponding dose in a human. The assumption for a human is 2 m2 for body surface. This number is under critical evaluation as some oncologists and surgeons argue that it should not be used as a standard in chemotherapy. However, the discussion on this subject is ongoing and it is still used as a standard.

  1. Of other foam carriers what was the rationale to choose bicarbonate-based foam carrier in this study?

Good point and important question. There was a prior study with unpublished data. Different carrier systems were tried, and in-vitro and post-mortem tissue studies indicated a bicarbonate carrier to be probably an acceptable solution and compromise.

  1. Blood parameters were measured after 30mins. Did authors measure blood parameters at any other time point like study midpoint?  

Thank you for this question. No further intraoperative blood parameters were taken.

  1. The abdominal expansion was measured in this article. Did the authors measure the kinetics of foam disintegration and time it took for complete foam disintegration?

Thank you for this. The kinetics of different carriers have been analyzed in a box model. This is currently unpublished data, and it is still under evaluation. The current experience is that the disintegration is a continuous process. Schubert et al. (Citation no 13) has published a curve on the disintegration of foam. But it is in a box model. There are many aspects how disintegration is taking place within the actual abdomen. We do not know exactly how it changes in the abdomen currently.

  1. Throughout the article the decimals are commas instead of dots.

Thank you we will go through them and correct accordingly.

  1. The authors did a good work in discussing their data well and highlighting the challenges of the study.

Thank you for your comment, your great questions and your appreciation.